# Correlates of Vaccine-Induced Protection against SARS-CoV-2

**DOI:** 10.3390/vaccines9030238

**Published:** 2021-03-10

**Authors:** Till Koch, Sibylle C. Mellinghoff, Parichehr Shamsrizi, Marylyn M. Addo, Christine Dahlke

**Affiliations:** 1Division of Infectious Diseases, 1st Department of Medicine, University Medical Center Hamburg Eppendorf, 20246 Hamburg, Germany; t.koch@uke.de (T.K.); sibylle.mellinghoff@uk-koeln.de (S.C.M.); p.shamsrizi@uke.de (P.S.); m.addo@uke.de (M.M.A.); 2German Centre for Infection Research (DZIF), Partner Site Hamburg-Lübeck-Borstel-Riems, 20359 Hamburg, Germany; 3Department for Clinical Immunology of Infectious Diseases, Bernhard Nocht Institute for Tropical Medicine, 20359 Hamburg, Germany; 4Excellence Centre for Medical Mycology (ECMM), 1st Department of Medicine, Faculty of Medicine and University Hospital Cologne, University of Cologne, 50937 Cologne, Germany; 5Cologne Excellence Cluster on Cellular Stress Responses in Aging-Associated Diseases (CECAD), Translational Research, Faculty of Medicine and University Hospital Cologne, University of Cologne, 50931 Cologne, Germany; 6German Centre for Infection Research (DZIF), Partner Site Bonn-Cologne, 50937 Cologne, Germany

**Keywords:** COVID-19, correlates of protection, immunogenicity, SARS-CoV-2, vaccine, pandemic

## Abstract

We are in the midst of a pandemic caused by the novel severe acute respiratory syndrome coronavirus 2 (SARS-CoV-2), which causes the coronavirus disease 2019 (COVID-19). SARS-CoV-2 has caused more than two million deaths after one year of the pandemic. The world is experiencing a deep economic recession. Safe and effective vaccines are needed to prevent further morbidity and mortality. Vaccine candidates against COVID-19 have been developed at an unprecedented speed, with more than 200 vaccine candidates currently under investigation. Among those, 20 candidates have entered the clinical Phase 3 to evaluate efficacy, and three have been approved by the European Medicines Agency. The aim of immunization is to act against infection, disease and/or transmission. However, the measurement of vaccine efficacy is challenging, as efficacy trials need to include large cohorts with verum and placebo cohorts. In the future, this will be even more challenging as further vaccine candidates will receive approval, an increasing number of humans will receive vaccinations and incidence might decrease. To evaluate novel and second-generation vaccine candidates, randomized placebo-controlled trials might not be appropriate anymore. Correlates of protection (CoP) could be an important tool to evaluate novel vaccine candidates, but vaccine-induced CoP have not been clearly defined for SARS-CoV-2 vaccines. In this review, we report on immunogenicity against natural SARS-CoV-2 infection, vaccine-induced immune responses and discuss immunological markers that can be linked to protection. By discussing the immunogenicity and efficacy of forerunner vaccines, we aim to give a comprehensive overview of possible efficacy measures and CoP.

## 1. Introduction

The ongoing coronavirus disease 2019 (COVID-19) pandemic has caused immense mortality and morbidity, and has also placed huge social and economic burdens on society. At the beginning of 2021, the global case count of infections with severe acute respiratory syndrome coronavirus 2 (SARS-CoV-2) has passed 110 million, with more than 2.5 million confirmed deaths due to the infection [1].

Vaccines can be a key element to limit viral spread. The search for an efficient vaccine has started in January 2020 and progressed at an unprecedented scope, both in the variety of vaccine platforms and in number of candidate vaccines under investigation. One year later, there are more than 250 vaccine candidates in development with 58 having progressed to clinical stages. Detailed lists can be found on various websites, e.g., by the World Health Organization (WHO) [2], London School of Hygiene & Tropical Medicine [3] or the New York Times [4].

In the beginning of 2021, three vaccine candidates have received regular licensure or emergency use authorization, including two mRNA-based and one non-replicating viral vector-based vaccine, in the United States (US), the European Union (EU) and the United Kingdom (UK). A second viral vector vaccine by Janssen (Johnson & Johnson) has been approved by the Food and Drug Administration (FDA) and is awaiting approval by the European Medicines Agency (EMA).

All four vaccines have announced efficacies ranging from 57% to 95% in Phase 3 trials in preventing COVID-19 [5,6,7,8]. Comparable efficacy was demonstrated in different populations in terms of gender, age or ethnicity, while efficacy and safety for populations such as pregnant women or children are still under investigation. Importantly, these four vaccines demonstrated mostly mild to moderate and transient reactogenicity [5,6,7,8]. Of note, it is yet to be determined if the current vaccine candidates are efficacious in reducing or even blocking transmission. A vaccine that confers sterilizing immunity or at least decreases the levels of viral shedding and subsequently infectiousness could significantly impact the containment of SARS-CoV-2.

The duration of immune response to SARS-CoV-2 vaccination remains to be investigated in the upcoming months. Antibody titers induced by natural coronavirus (CoV)-infection have been reported to wane over time. Specifically, human challenge models in past decades have demonstrated the possibility of re-infection with two common cold coronaviruses hCoV-229E and hCoV-OC43 [9,10]. An infection of SARS-CoV-1 or the middle east respiratory syndrome (MERS)-CoV revealed antibody responses that gradually decline over time [11,12]. The current limited data for SARS-CoV-2 infection indicates that humoral immune response persists for at least several months [13,14]. In the light of emerging SARS-CoV-2 variants, most studies assessed protection against the prototype SARS-CoV-2 sequence B.1 or the D614G variant. Whether humoral responses to SARS-CoV-2 may prevent re-infection with novel variants like B1.1.7, B1.315 or P.1 remain unclear to date. First analyses however show reduced neutralization of convalescent plasma in vitro concerning the P.1 or B1.315 variant [15].

Efficacy for the recently approved vaccines has been assessed in randomized placebo-controlled clinical trials (RCT). These trials are considered the gold-standard in clinical research, as they limit the potential for bias in data collection and deliver the highest level of scientific evidence. However, these trials are extremely demanding in terms of resources of any kind. Their implementation may become even more difficult for novel and second-generation vaccine candidates once infection incidence has decreased, or vaccines against SARS-CoV-2 have been properly licensed. With regard to potentially upcoming mutations in the SARS-CoV-2 genome resulting in an escape from human neutralizing antibody (nAb) responses, the introduction of adapted vaccines may be essential to halt the pandemic. RCT might prove to be no longer feasible due to time, cost or ethical reasons. A reasonable alternative would be the measurement of immunity. An immune response that is responsible for and statistically interrelated with protection is named correlate of protection (CoP). The use of CoP may allow the prediction of clinical outcomes (protection against disease or infection) after vaccination or natural infection [16].

CoP can be divided into mechanistic CoP (mCoP), which are causal for protection, and non-mechanistic CoP (nCoP), which are a predictor of protection without being its causal agent [17]. A surrogate is an immune response that substitutes for the true immunological correlate of protection. The identification and measurement of CoP are often challenging. A potential surrogate endpoint for a SARS-CoV-2 vaccine would most likely depend on the characteristics of the vaccine including antigen, method of delivery and method of antigen presentation utilized by the vaccine. 

In this review, we delineate briefly the SARS-CoV-2 induced immune responses and review results of peer-reviewed publications on four forerunner COVID-19 vaccine candidates. Finally, we will discuss the use of correlates of vaccine-induced protection against SARS-CoV-2, urgently needed for the advancement of novel and second-generation vaccine candidates.

## 2. Immune Response to SARS-CoV-2

SARS-CoV-2 infection can induce SARS-CoV-2 specific antibodies, CD4+ and CD8+ T-cells (Figure 1 and Figure 2), which target the four structural proteins spike (S), matrix (M), nucleocapsid (N) and envelope (E) as well as to some extent the non-structural proteins [18]. It is assumed that disease severity may critically impact the quality, quantity and duration of immune responses to SARS-CoV-2. Further, numerous reports have discussed the role of nAb and T-cell responses on the severity of the clinical course in COVID-19 [19,20], but there are still knowledge gaps on the broadness, robustness and durability of immune responses. Data on SARS-CoV-2 immunity are emerging rapidly and updates of novel findings are reported by several webpages like the UK biobank report [21].

### 2.1. Humoral Response

Antibody affinity and avidity are both quality parameters to effectively block viral spread. SARS-CoV-2 infection generally leads to antibody responses against N and S. SARS-CoV-2 expresses the S glycoprotein on its surface. The protein consists of the two subunits S1 (N-terminal) and S2 (C-terminal) [18]. The S1 subunit facilitates binding to its receptor angiotensin-converting enzyme 2 (ACE2) [22,23] via the receptor-binding domain (RBD), while S2 facilitates membrane fusion. Antibodies targeting the RBD are considered to be important for viral neutralization, since binding to the virus precludes its attachment and the entry into the host cell.

Patients recovered from COVID-19 have been shown to maintain S1- and S2-specific antibodies with neutralizing activity against SARS-CoV-2, which correlates positively with disease severity [24]. The majority of patients show SARS-CoV-2-specific immunoglobulin (Ig)A, IgG and IgM antibody responses shortly after infection that persist for weeks to months (Figure 2) [25,26,27]. Several factors, upfront age and sex, might influence antibody kinetics [28,29]. The serological IgA response occurs first and peaks early [30]. In particular, dimeric IgA, the primary nasopharyngeal antibody compound, seem to be potent compared to monomers [31] and might be an important marker for protection and vaccine efficacy. Secretion of SARS-CoV-2 specific IgM and IgG is reported to occur almost simultaneously 7 to 14 days after symptom onset [20,32]. Individuals with severe illness show higher levels of nucleocapsid-specific IgM and IgG, which are unlikely to have direct neutralizing effects towards the virus, but rather might exert protection by indirect, fragment crystallizable (Fc)-mediated effector functions. It has recently been described that patients with severe COVID-19 tend to have a specific serologic signature which includes an increased level of IgG antibodies with a specific afucosylated Fc part [33,34]. This mediates an enhanced FcγRIIIa interaction, leading to increased cytokine secretion and immune-mediated pathology.

The vast majority of SARS-CoV-2 infected individuals seroconvert, with some studies reporting unchanged antibody titers for several months past infection [13]. Other seroprevalence studies report that humoral immunity is not long-lasting, especially in individuals with mild disease, describing a decline in IgG after two to four months [35,36,37]. Long-term studies are so far not available for SARS-CoV-2, but observation from other coronaviruses show waning antibody titers over time [38]. Considering that IgG titers decline even further in asymptomatic than in symptomatic patients in the convalescent phase [39], implications for serological observations and immunization strategies need to be evaluated.

There is evidence that nAb are good markers for protective immunity against re-infection. In a prophylactic or therapeutic setting, neutralizing monoclonal antibodies isolated from COVID-19 patients have been shown to inhibit SARS-CoV-2 infection in animal models [40,41,42]. 

### 2.2. Cellular Immune Response

T-cells are key players in coordinating anti-viral immune responses. They induce killing of infected cells or mediate humoral responses. Current data underscore that not all patients may develop a protective humoral immune response, but still generate robust T-cell responses. Notably, data from SARS-CoV-1 infections have indicated that antibodies often wane 1–2 years after infection, while T-cell responses can last up to 17 years [43]. 

In SARS-CoV-2 infection, T-cell activity is associated with lower disease severity, indicating that T-cells are important for control and resolution of primary SARS-CoV-2 infection [19]. Further, pre-existing, cross-reactive T-cells might accelerate virus clearance to SARS-CoV-2 [44,45], but their importance remains unclear. There is evidence that a balanced T-cell response can prevent or dampen the course of COVID-19, while a delayed or inadequate response may lead to an uncoordinated and inefficient virus control with subsequent exacerbated tissue damage [46]. Monocytes can then be activated and produce high levels of proinflammatory cytokines like interleukin 6 (IL-6) [47]. Given the large number of T helper (Th)1 cells and inflammatory monocytes in bronchoalveolar lavage specimens and lung biopsies from COVID-19 patients with severe illness [48], an excessive cellular immune response may substantially impact functional pulmonary disability by damaging pulmonary microcirculation.

Future studies need to evaluate the role of cellular immunity in long-term protection, specifically of tissue-resident populations. T-cellular immune response to viral infection is mediated by interferon (IFN) with a major role of type I IFN [49]. IFN-stimulated genes (ISGs) have been shown to be significantly reduced in critical COVID-19 patients compared with patients that experienced mild to moderate infection [50]. An impaired IFN I-phenotype in patients with severe COVID-19 suggests that SARS-CoV-2 is capable of inducing efficient mechanisms to dampen or delay host IFN production potentially contributing to immunopathology [49,50,51,52]. Further, a recent study has shown IFN-specific auto-antibodies in patients with severe COVID-19 and hypothesized that their production contributes to IFN-impairment [53]. Autoantibodies against one or more cytokines have been reported in different conditions, but their biological role needs yet to be defined [49]. In addition, host factors such as comorbidities may negatively affect IFN production.

While individuals who have recovered from mild COVID-19 disease sometimes lack detectable antibody responses, T-cell responses could often be identified [45]. T-cells (CD4+ and CD8+) respond within the first two weeks after onset of symptoms [18], typically by Th1 activation [54,55]. During the acute phase of infection, they display an activated and cytotoxic phenotype. In the following convalescent phase, virus-specific T-cells can change towards a memory phenotype with CD4+ as well as CD8+ T-cells expressing IFNγ, interleukin-2 (IL-2) and/or tumor necrosis factor α (TNFα) [45]. In mild disease, T-cell proportion was attributable to CD8+ T-cells, while severe cases show a relatively high frequency of SARS-CoV-2 specific CD4+ T-cells. Spike specific CD4+ T-cell responses were more abundant in severe than mild cases. Whether the increase of CD4+ T-cells in patients with severe disease reflects an increased antigenic load driving stronger immune responses remains to be elucidated. With regard to MERS, memory CD4+ T-cells were associated with survival in humans [56]. In animal models testing SARS-CoV-1 and MERS-CoV infection, T-cell responses were critical for clearance of infection [57,58].

Persistent memory populations can rapidly expand on rechallenge. Considering reinfection, broad cellular responses to a number of different SARS-CoV-2 specific peptides may facilitate protection [54]. Evaluations of T-cell responses to a number of specific peptides revealed a multi-specific response to such proteins.

In conclusion, several factors such as age, obesity, or sex can influence the risk to develop severe COVID-19 [59]. Coordinated T-cell and antibody responses appear to be protective, while uncoordinated responses fail to control or even promote disease.

## 3. Immune Response to SARS-CoV-2 Vaccines

Vaccines against COVID-19 will play a pivotal role for limiting the pandemic. The majority of current vaccine candidates utilize S (or parts thereof like RBD), due to its crucial role in mediating viral entry into cells.

Three vaccine candidates (BioNTech/Pfizer, Moderna & AstraZeneca, Cambridge, UK) have received approval by the EMA, and one is under investigation by the rolling review (Janssen, Belce, Belgium). We here focus on these four forerunner vaccine candidates and delineate their immunogenicity data gained in the framework of Phase 1–3 trials.

The vaccine candidate AZD1222 (Astra Zeneca) is based on a replication deficient chimpanzee adenoviral vector (ChAdOx1) encoding the S protein [7,60]. In a Phase 2/3 trial, volunteers received 5 × 10^10^ virus particles (vp) as a standard dose for both prime and boost immunization. In addition, a small cohort received a lower dose (2.5 × 10^10^ vp) as prime immunization. The different study sites used a boost interval that ranged between 4 to 12 weeks. Efficacy in prevention of COVID-19 varied from 62% to 90%, with the variance likely due to heterogeneity in dosing, prime-boost intervals and diversity of study populations. While further studies are ongoing, the vaccine has been approved in the UK with an admitted variance in prime-boost interval between 4 and 12 weeks. The EU has just recently approved the vaccine [61].

Immunogenicity data were reported from a Phase 1/2 trial conducted in the UK [62]. Here, anti-spike IgG responses were observed 28 days after prime with a further increase of titers following boost immunization. Neutralizing antibodies were observed in 91% of participants after prime and 100% after boost immunization in an MNA80 (live SARS-CoV-2 microneutralization) assay. Cellular immunity was also induced by prime and boost immunization, while the boost did not significantly impact the IFN-γ responses (measured with IFN-γ enzyme-linked immunosorbent spot (ELISpot)).

Ad26.COV2.S is another adenovirus-vectored vaccine developed by Janssen. The vaccine is based on a recombinant, replication deficient adenovirus (Ad26) encoding a full-length and stabilized spike protein [63]. In a placebo-controlled phase 1/2a trial, a low dose (5 × 10^10^ vp) and high dose (1 × 10^11^ vp) were evaluated as single dose or combined with a booster dose after 56 days. The single dose regimen showed promising results in immunogenicity analyses that warrant further evaluation. After a single vaccination, nAb titers were detected in 90% or more of participants on day 29 and 100% on day 57. Spike-binding antibodies as measured by ELISA correlated well with nAb titers, especially in younger adults. CD4+ T-cell responses on day 14 were induced in 76%–83% of the participants (depending on the dose) with a trend toward type 1 helper T-cells. CD8+ T-cell responses were induced in 51%–64% of participants. Cellular responses were generally lower in higher age groups.

Preclinical data revealed that a single injection resulted in complete protection in lower and upper respiratory tract in rhesus macaques [64]. Clinical data from the Phase 1/2 study also showed induction of T-cell responses and importantly nAb in all participants after a single dose of Ad26.COV2.S (5 × 10^10^ vp). Janssen therefore initiated a Phase 3 study to evaluate the efficacy of a single dose regimen (clinicaltrials.gov: NCT04505722). By the end of January 2021 a press release was published [8], announcing that the single-shot regime provides efficacy of 57%–72% against moderate to severe COVID-19 and is 85% effective in preventing severe COVID-19. In addition, the efficacy of a two-dose regimen is currently investigated in a parallel trial (clinicaltrials.gov: NCT04614948).

RNA vaccines have been the forerunners in vaccine development against COVID-19. The two RNA-based vaccines developed by BioNTech/Pfizer and Moderna have already received approval from EMA and FDA in December 2020/January 2021. Both are lipid nanoparticle (LNP) formulated nucleoside-modified mRNAs, encoding the stabilized prefusion SARS-CoV-2 spike protein that are administered intramuscularly. Pfizer/BioNTech’s BNT162b2 contains 30 µg of RNA and is administered 21 days apart [5], while Moderna’s mRNA127 contains 100 µg of RNA and is administered by an interval of 28 days [6].

Immunization with BNT162b2 induced binding IgG antibodies against S1 following a single injection, while nAb were detectable in the majority of vaccinees earliest at day 28, seven days following the boost immunization [65]. CD4+ and CD8+ T-cell responses on day 29 were induced in 94.1% (32 out of 34) and 91.9% (34 out of 37) of participants, respectively. T-cell analysis was conducted by an ex vivo IFNy ELISpot assay using stimulation of either CD4+ or CD8+ cells by overlapping peptide pools covering the spike protein. Vaccine efficacy was analyzed in the Phase 3 trial including over 40,000 volunteers that received either BNT162b2 or placebo. Here, the case split of 8 versus 162 COVID-19 cases in the verum and the placebo arm demonstrated a 95% efficacy in prevention of COVID-19 [5].

The mRNA-1273 vaccine by Moderna showed immune responses after the prime injection, with a booster injection resulting in increased titers of both binding and nAb in all participants evaluated in the Phase 1 trial [66]. T-cell responses were analyzed using two pools covering S1 and S2. Here, a Th1-dominant CD4 T-cell response was observed, while CD8+ T-cell responses were low when analyzing responses by intracellular cytokine-staining assay using flowcytometry. In the Phase 3 efficacy trial, over 30,000 volunteers were enrolled and received either mRNA-1273 or placebo. Vaccine efficacy in prevention of COVID-19 was 94.1% with a case split of 11 versus 185 participants in the vaccine and placebo group, respectively [6].

Details on these four forerunner vaccines are listed in Table 1.

Further vaccine candidates are expected to be approved soon, most of them will be administered intramuscularly. While those generally induce systemic immune responses with dominant IgG responses, natural infection induces both systemic and mucosal immune responses [67,68]. The induction of mucosal immune response in the upper respiratory tract generally leads to secretion of secretory IgA, which can be an important factor to induce sterilizing immunity preventing infection and virus transmission [68]. A vaccine candidate that induces mucosal immune response in the upper respiratory tract and thereby potentially sterilizing immunity would be preferable. It has been shown that e.g., application of viral vectors intranasally can lead to strong mucosal immune responses as well as an IgG response [68]. To date, six intranasal and three oral vaccine candidates are in clinical Phase 1 or 2 trials [2]. While data from clinical trials have not yet been published, preclinical studies suggest the induction of mucosal immunity [69]. First results from clinical trials of an oral vaccine candidate by Vaxart Inc. have recently been announced [70].

## 4. CoP in Vaccine Development against SARS-CoV-2

### 4.1. Study Designs to Evaluate Vaccine Efficacy

The development of vaccine candidates against COVID-19 was tremendously fast with two vaccines approved after less than a year. Current Phase 3 trials test or already tested the efficacy of their COVID-19 vaccine in randomized double-blind placebo control (RDBPC) trials as this design represents the gold standard. However, with approved vaccine candidates it is likely that strategies aiming to evaluate vaccine efficacy need to be adapted. RDBPC are becoming inappropriate from a scientific and ethical perspective given the increasing number of vaccinated individuals and the likelihood of reduced SARS-CoV-2 incidence. There are different strategies that are conceivable to evaluate efficacy of COVID-19 vaccines in the future.

Continue a blinded follow-up trial until participants become eligible for vaccination in national programs and/or when they wish to receive a vaccine that is nationally available. As an alternative, continue placebo-controlled trials including only individuals that are not eligible for vaccination in national programs.Conduct head-to-head efficacy trials. A notable disadvantage is the need to include a large number of participants and the possibly long duration of the study. Yet, head-to-head comparisons of vaccines can be important to understand specific CoP by different vaccines.Compare efficacy in human challenge studies (HCS). While UK has recently started HCS, there are still debates elsewhere whether these trials are ethically justifiable at this point in the pandemic.Evaluate new vaccine candidates based on established immunological CoP. Here, we need to define whether we are interested in protection against infection or against disease. Using a specific immunological threshold linked to protection can dramatically accelerate screening and selection or de-selection of novel vaccine candidates.

The evaluation of new vaccine candidates and adapted candidates to new SARS-CoV-2 variants emphasizes the need to adapt study designs.

### 4.2. General Considerations

For several reasons including costs, time and ethical considerations, the use of clinical endpoints alone is unfavorable. The use of CoP might be critical to rapidly respond with new vaccine candidates to emerging variants. It is therefore important to identify and use simple substitute endpoints instead of clinical endpoints to evaluate vaccine efficacy. An immune CoP is an immunological marker that reliably predicts protection against disease or infection after natural infection or vaccination. While infection might be blocked by nAb, disease progression is likely to be influenced by cellular immunity. Considering the designs outlined above, the use of immunological CoP would be a very effective, rapid and ethically justifiable way to evaluate novel vaccine candidates. However, CoP after infection might differ from CoP after vaccination and commonalities between natural immunity after SARS-CoV-2 infection versus vaccine-induced immunity are so far unknown.

For many vaccines, antibodies are the main driver for protection and are known to be a CoP [71]. They need to be functional, but not necessarily neutralizing. For example, Hepatitis B vaccinees showed detectable memory B cells despite non-protective antibody levels; and within an HIV-1 vaccine trial, IgG3 antibody dependent cell-mediated cytotoxicity antibodies (ADCC), but not nAb correlated with protection [72]. There are also vaccines with cellular immune responses as immune correlates, e.g., against Varicella Zoster virus [71,72,73].

It is important to understand that vaccination generally induces several immune markers, but only some induced factors might be required to enable protection. The complexity of interaction and the identification of relevant single or combined CoP is challenging. Interrelationships between vaccination, immune responses and clinical endpoints need to be considered as well as timing of CoP measurements. Antibody kinetics during the course of SARS-CoV-2 infection indicate an antibody (IgG) peak approximately three weeks after onset of infection (Figure 2). In contrast, vaccinees are exposed twice to immune stimulation by the vaccine and a significant increase in antibody titers is detected at day 42 after first vaccination (Table 1). Kinetics of cellular immune responses as well as antibody kinetics in those patients that receive a delayed second dose remain to be investigated and best timing of CoP measurement is yet to be determined.

Good examples to delineate the complexity of identifying CoP are measles [74] and poxviruses vaccines. In case of the measles vaccine, a specific range of binding antibodies indicate protection against measles disease in most cases, but not always against infection. In addition, animal models revealed that CD8+ T-cells are required to suppress viremia [75]. The poxvirus vaccine leads to humoral and cell-mediated responses. While nAb are required for protection, anti-poxvirus T-cells prevent a severe clinical course in case of reinfection when antibody titers have already declined [76,77,78]. Of note, protection against diseases like malaria or tuberculosis is mainly T-cell mediated [71,72]. However, measurement of T-cell responses is complex due to the different T-cell subsets with various functions like IFNγ, IL-2, TNFα or granzyme B (GrzB). Further, T follicular helper (Tfh) cells may also be linked with protection [79].

The SARS-CoV-2 vaccine landscape yields various candidate vaccines that have been developed and tested, including nucleic acid vaccines, inactivated virus vaccines, live-attenuated vaccines, protein or peptide subunit vaccines and viral-vectored vaccines [68,80]. Each vaccine type has specific advantages and disadvantages; and each approach might mediate specific vaccine-induced mechanism of protection.

### 4.3. Study Designs to Evaluate CoP

The identification of immunological correlates requires the combination of multiple data sources. Vaccine efficacy studies, natural infection studies and passive immunization studies can significantly contribute to the understanding of mechanisms of protection against SARS-CoV-2 infection or COVID-19 disease. To identify CoP (Figure 3) within the framework of vaccine pre-clinical and clinical trials, well-designed methods for measuring and assessing CoP in relation to efficacy and effectiveness are required. Options are the analysis of protected versus unprotected vaccines in efficacy trials, using HCS or animal studies, including immunodeficiency models.

Efficacy trials are ideal to investigate the differences from vaccinees that were protected versus vaccinees that failed protection. Large-scale implementations of first-generation vaccines against COVID-19 will provide an important opportunity to collect data on immunologic correlates of protective responses. Such data can facilitate the licensure of many second-generation vaccines.
Randomized-controlled trials. RCT provide an ideal setting to assess vaccine efficacy, the association between vaccination and immune markers and the association between markers, protection and clinical endpoints. As an example, an RCT testing *Haemophilus influenzae* type b (Hib) polysaccharide conjugate vaccine investigated associations between vaccination and clinical infection, observing 95% (95% CI: 72%–99%) protection with an antibody concentration of 0.15 μg/mL [81].Immunogenicity trials (Phase 2 trials). Within these trials, individuals receive a vaccine at different doses or a placebo and their immune responses are compared. These trials include a smaller set of participants but allow comprehensive analysis of immunogenicity. As an example, Meningococcal C conjugate vaccine was licensed in the United Kingdom. A study compared serum bactericidal assay titers induced by the new vaccine to those induced by a licensed serogroup C polysaccharide vaccine, which demonstrated direct evidence of efficacy and accepted correlates of protection [82].Passive immunization studies. Here, specific IgGs are administrated and evaluated whether they can protect against disease.HCS. Following vaccination, volunteers are challenged with the pathogen. Here, a direct association of immune markers and protection can be made. However, there are several caveats, e.g., HCS are conducted with young and healthy volunteers and do not equally represent the whole populations like elderly, children or immunocompromised individuals and thereby lack information on specific target populations.Observational studies. They imply studies with only passive observation of groups or populations, with no controlled intervention. Observational studies include cohort or natural history studies.Case control studies. The case–control approach has been used to compare levels of immune markers prior to disease among individuals who did or did not develop the clinical outcome of interest. The blood samples must have been collected prior to disease onset, and preferably prior to pathogen exposure.

Knowledge on natural immunity, in particular the understanding of protection versus insufficient protection from re-infection, provides in-depth insight into protective mechanisms and possible measurable correlates. Large cross-sectional studies suggest that SARS-CoV-2-specific CD4+ and CD8+ T-cells in coordination with nAb can generate protective immunity as observed in most COVID-19 cases in humans and non-human primate (NHP). Circulating nAb have the potential to serve as a significant correlate of protection against disease in humans. Although case studies may lack power, studies of so far rare re-infections or small outbreak scenario reports are important for understanding immune protection. For example, a study reported from an outbreak of SARS-CoV-2 on a fishing vessel showed a high attack rate of 85% [83]. Three crew members with nAb titers showed no evidence of re-infection and did not experience any symptoms during the viral outbreak. This report provides clinical evidence that anti-SARS-CoV-2 nAb protect against re-infection.

Passive immunization through monoclonal antibodies is currently under investigation. First evidence on prevention of severe disease in high-risk individuals in an outpatient setting [84,85] have led the FDA to grant emergency use authorization to two products, namely REGN-COV2 and LY-CoV555.

### 4.4. Animal Studies

In the absence of human data, results of animal models may help to identify potential CoP. Currently, animal models including mouse, hamster, ferret and NHP have been characterized and tested [86]. The hamster model can mimic severe disease as seen in a small subset of infected humans, while the NHP model rather reflects the mild to moderate course of disease seen in the majority of human cases. All forerunner vaccine candidates have been tested in NHP models [68]. With regard to natural infection, NHP revealed protection in case of re-infection with previous SARS-CoV-2 infection [87,88]. A comparison of six vaccine candidates (Sinovac, Sinopharm, AstraZeneca, Janssen, Moderna, Novavax) outlines that in vaccinated and challenged NHP, lungs were completely protected following vaccination across all candidates. All animals developed nAb. Vaccine-mediated responses showed partial/complete protection of upper and lower respiratory tracts. To block transmission, it might be pivotal to protect specifically the upper respiratory tracts. While upper respiratory tracts were only completely protected following vaccination with Ad26.CoV2.S (Janssen) and NVX-CoV2373 (Novavax), data from NHP needs to be compared with human trials. Whether vaccines can protect against virus transmission in humans is currently under investigation.

Animal and human data (i.e., data from NHP models and mRNA-based vaccines’ clinical trials) provide strong evidence that nAb can serve as a mechanistic CoP and that even low levels are protective regarding disease. One NHP study tested a DNA vaccine candidate and found that nAb, but not T-cells, correlated with protection [89]. In another study on NHP, purified antibodies from convalescent macaques protected naïve animals against SARS-CoV-2 infection of both upper and lower airways in a dose-dependent fashion [90]. In the same study, the depletion of CD8+ T-cells resulted in reduced protection from infection in the lower and diminished protection in the upper respiratory tract, indicating a role of CD8+ cellular immune response in protection from infection. Just like the DNA vaccine, an Ad26-based vaccine also induced nAb in rhesus macaques that strongly correlated with a reduction of viral loads [64,89], supporting the assumption that nAbs are a measurable CoP. In addition, inactivated virus vaccines and mRNA vaccines induced nAbs and conferred protection in macaques, supporting this hypothesis [91,92]. With regard to T-cells, only AstraZeneca, Moderna and Janssen investigated cellular responses. T-cell responses were detectable for the AstraZeneca and Moderna vaccine and at low levels for the Janssen vaccine candidate in animal models [68]. T-cell responses might be a critical parameter for moderate versus severe disease progression.

If nAb prove to be a robust CoP over multiple studies in both NHP and humans, this parameter would be one valuable measure for clinical development and validation of future SARS-CoV-2 vaccines.

### 4.5. Methods for Evaluation of CoP

The identification of CoP and finding a specific cutoff level is complex and often challenging. As described elsewhere [93], statistical methods are needed to assess cutoff levels, i.e., the threshold method, continuous method based on case–cohort studies and receiver operating characteristic (ROC) curve method by using case–control design.

While a hypothetical relationship between nAb titers and clinical protection may exist, identification is challenging due to heterogeneity of immune response and exposure dose. Underlying causes of heterogeneity are the complexity of immune responses like T-cell mechanisms, non-neutralizing antibodies, differences to age or sex, as well as previous exposure to other CoV. The role of non-neutralizing antibodies, that mediate their function via Fc receptors, in SARS-CoV-2 infection, needs to be further investigated, analyzing complement-dependent cytotoxicity (CDC), ADCC or antibody-dependent phagocytosis (ADP) (Figure 3D). In addition, mucosal immunity might also play a pivotal role in protection.

To evaluate immunogenicity to SARS-CoV-2 following infection or vaccination has not been conducted in a standardized way. There are multiple assays performed and comparisons are difficult to make, although efforts are underway to standardize assays on SARS-CoV-2, e.g., by the WHO [94]. Current efforts aim to find robust and valid surrogate assays. ELISA tests identify binding of IgG, IgM or IgA to purified proteins of SARS-CoV-2 instead of live virus. Here, assays against S or specifically the RBD region are often used. In addition, surrogate virus neutralization tests (sVNT) have been developed to detect potential nAB, often those that prevent interaction of RBD with ACE2. Beside binding antibodies, neutralization tests determine the functional ability of antibodies to prevent virus infection in vitro. The tests involve incubating serum or plasma with live virus followed by infection and incubation of cells. There are two types of tests: Virus neutralization tests (VNT), such as the plaque-reduction neutralization test (PRNT) and microneutralization. They generally use a SARS-CoV-2 virus from a clinical isolate. Pseudovirus neutralization tests (pVNT) use recombinant pseudoviruses (like vesicular stomatitis virus (VSV)) that incorporate the S protein of SARS-CoV-2. In comparison, T-cell responses are currently analyzed by ELISpot assays, but assays that identify T-cell responses without conducting time-intense and operator-dependent ELISpot assays are under investigation.

Due to the heterogeneity of immune responses and range of vaccines being designed and developed, there is a need to define clear and international standards to assess and interpret the results of comparative immunogenicity trials. The WHO and Coalition for Epidemic Preparedness Innovation (CEPI) are currently working towards a harmonized assessment of the immune responses generated by different COVID-19 vaccine candidates. To reduce interlaboratory variability, international standards of virus strains are available for testing nAb functions.

## 5. Conclusions

Immunological correlates are critical to facilitate the evaluation on vaccine efficacy. So far, we do not fully understand vaccine-induced immunogenicity and the mechanisms that protect against infection, disease or fatal COVID-19. Further, we need to understand how virus shedding can be blocked or at least reduced by vaccination and how long protection lasts following infection or vaccination. Another aspect is the knowledge on protection in certain populations like children, pregnant individuals or elderly. In addition, protection against emerging variants of SARS-CoV-2 should be monitored.

Multiple safe and effective vaccines are needed to control SARS-CoV-2 in the long run and hopefully end the pandemic. To achieve this goal, novel vaccine candidates are likely needed in addition to approved vaccines. The use of reliable immune markers like antibody threshold levels that correlate with protection from SARS-CoV-2 may enable the assessment of vaccine efficacy in the absence of data from placebo-controlled trials and thereby speed up the evaluation and approval of vaccines.

## Figures and Tables

**Figure 1 vaccines-09-00238-f001:**
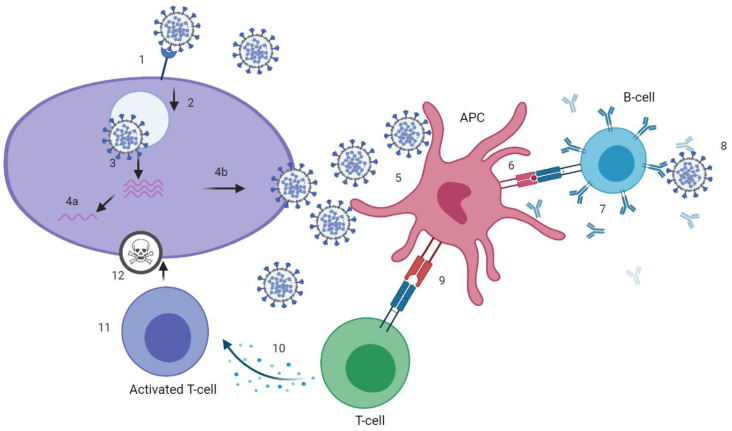
Immune response to severe acute respiratory syndrome coronavirus 2 (SARS-CoV-2). (1) attachment of SARS-CoV-2 virus to host cell via angiotensin-converting enzyme 2 (ACE2)-receptor. (2) cell-entry. (3) membrane fusion and RNA-release into host cell. (4a) presentation of RNA and viral proteins to Toll-like receptor (TLR) and activation of innate immune response. (4b) assembly of virions. (5) uptake of virus by antigen presenting cell (APC). (6) presentation of antigens, including epitopes, to B-cell receptor (BCR). (7) production of binding and neutralizing antibodies by B-cells that, ideally, (8) neutralize the virus. (9) presentation of antigens, including epitopes, to T-cell receptor (TCR). (10) Activation of T helper (Th) cells and production of cytokines, that, recognized by (11) cytotoxic T-cells, (12) kill the virus. Graphic created at biorender.com.

**Figure 2 vaccines-09-00238-f002:**
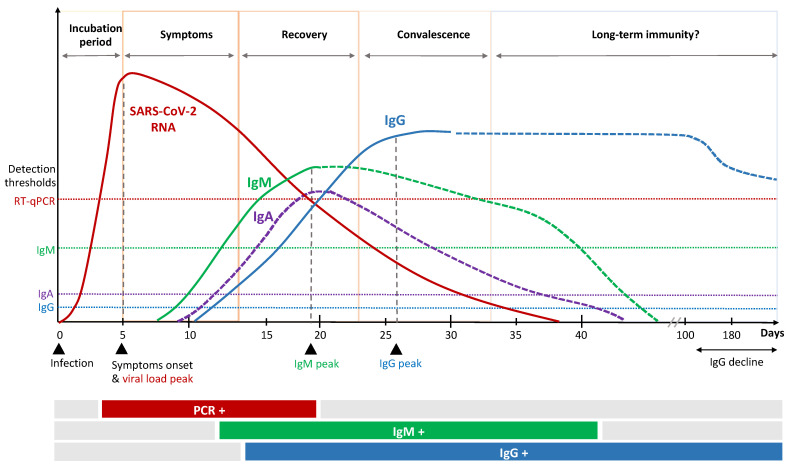
Kinetics of immune response to SARS-CoV-2 infection.

**Figure 3 vaccines-09-00238-f003:**
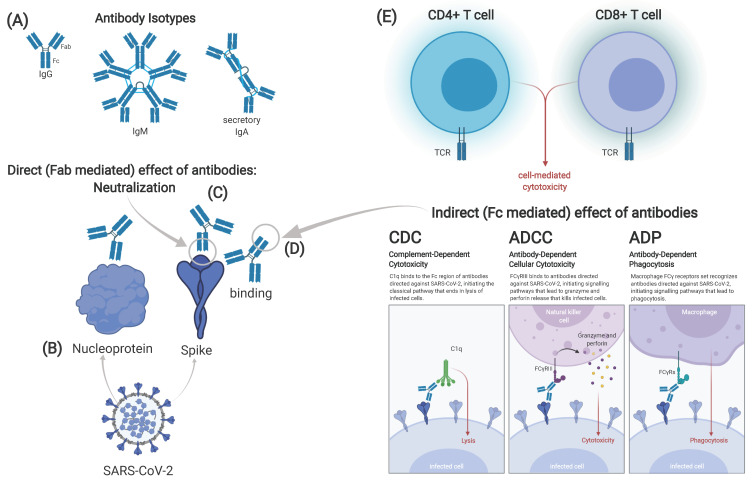
Possible correlates of protection against SARS-CoV-2. Different isotypes of antibodies can be identified after vaccination or infection. Immunoglobulin (Ig)G and IgM are found in serum, while secretory IgA in its dimeric form is found on mucous membranes (**A**). Antibodies can target either the spike or other viral proteins, e.g., the nucleoprotein (**B**). Effect of antibodies can either be direct, mediated through the fragment antigen binding (Fab) part of the antibody (**C**) or indirect, mediated by the fragment crystallizable (Fc) part (**D**). The latter includes complement-dependent cytotoxicity (CDC), antibody dependent cell-mediated cytotoxicity antibodies (ADCC) and antibody-dependent phagocytosis (ADP). Cellular immune responses can be divided in CD4+ and CD8+ T-cell responses (**E**). Correlates of protection may differ in infection and vaccination, maybe even between different types of vaccines. Graphic created at biorender.com including adapted template from Daniela Rothschild Rodriguez.

**Table 1 vaccines-09-00238-t001:** Overview of forerunner vaccine candidates.

Company	Vaccine (Type)	Trial (Ref) and NCT	Humoral Response (Geometric Mean Titer)			Cellular Response (SARS-CoV-2 Specific)	
			After 1st Dose	After 2nd Dose		CD4	CD8
Pfizer	BNT162b2 (mRNA expressing spike protein)	Phase1/2 [53] NCT04380701	1:312 (day 7) ^a^	1:181 (day 85) ^a^		CD4 T cells in 37/37 pts. (day 7 after boost), in 30/34 *de novo* response compared to baseline; Th1 > Th2	SARS-CoV-2 specific CD8 T cells in 34/37 pts. (91.9%)
Moderna	mRNA-1273 (mRNA expressing spike protein)	Phase 1 (adults 18 to 55 years) [54] NCT04283461	1:4 (day 1) ^b^	1:654.3 ^b^ (day 43)		CD4 T-cell response Th1 > Th2	CD8 T-cell response at low level only after 2nd dose
		Phase 1 (adults 56 to 70 years and ≥71 years) [67] NCT04283461	n/a	Age 56 to 70		CD4 T-cell response Th1 > Th2 in both age groups	CD8 T-cell response at low level only after 2nd dose
				1:402 ^c^ and 1:878 ^d^ (day 43)			
				Age ≥ 71			
				1:317 ^c^ and 1:317 ^d^ (day 43)			
		Phase 3 (interim analysis) [6,68] NCT04470427	n/a	Age 18 to 55		CD4 T-cell response Th1 > Th2 in all age groups	n/a
				1:182 ^c^ and 1: 430 ^d^ (day 119)			
				Age 56 to 70			
				1: 167 ^c^ and 1:269 ^d^ (day 119)			
				Age ≥ 71			
				1:109 ^c^ and 1:165 ^d^ (day 119)			
AstraZeneca	ChAdOx1 nCoV-19 (non-replicating chimpanzee Ad. expressing spike protein)	Phase 2/3 [48] NCT04400838	n/a	LD/SD (day 42)	SD/SD (day 42)	Only available for subgroup (age 18 to 55 years, SD): IFN-γ ELISpot response against SARS-CoV-2 spike protein peaked 14 days after the prime vaccination	n/a
				Age 18 to 55			
				1:161 ^e^	1:193 ^e^		
				Age 56 to 69			
				1:143 ^e^	1:144 ^e^		
				Age ≥ 70			
				1:150 ^e^	1:161 ^e^		
Janssen	Ad26.COV2 (recombinant, replication-incompetent adenovirus serotype 26 (Ad26) vector encoding a full-length and stabilized SARS-CoV-2 spike (S) protein)	Phase 1-2a [51] NCT04436276	1: 310 (day 57, age 18 to 55 ^f^)	n/a		Th1 response to S peptides in - 76% (of low-dose recipients - 83% (of high-dose recipients Th1 > Th2	Response detected in - 51% of participants in low-dose group ^g^ - 64% in high-dose group ^g^

^a^ Result only reported for 30 µg dose; based on microneutralization assay with a SARS-CoV-2 reporter virus, 50% neutralization titer (VNT50) as readout. ^b^ Result only reported for 100 µg dose; based on PRNT80 with authentic SARS-CoV-2. ^c^ Based on ID50 pseudovirus neutralization assay. ^d^ Based on PRNT80 with authentic SARS-CoV-2. ^e^ Based on live SARS-CoV-2 microneutralization assay (MNA80). ^f^ Result only reported for 5 × 10^10^ virus particle single dose. ^g^ Identified by expression of INF-γ or IL-2 cytokines on S-peptide stimulation.

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
