# Peer review of "Correlates of Vaccine-Induced Protection against SARS-CoV-2"

_vaccines, 2021, doi:10.3390/vaccines9030238_

Round 1

Reviewer 1 Report

The subject is very timely for a review, however, the authors fail to address some key issues. To name a few: epitope mapping and HLA restriction of antibody responses to SARS-CoV2; the presence of natural antibodies in non-infected subjects; local and systemic mucosal responses and their memory (oral boosters); abnormal antibodies (see a-fucosylated IgG, their binding to activating monocyte Fc receptors); and finally, poor type I IFN responses and/or presence of antibodies to IFN type I. The coverage of vaccine trials and their published correlates is adequate. The use of possible standardized tests is not convincingly advanced, no examples are offered. There are problems with the English language (e.g. lines 190. 305 (an), 343 and following 420). Some abbreviations are not explained (nAB line 110, then newly in line 416, are they the same?), NHP line 338,  Missing references needed for HepB (line 101), measles (ref 17 is generic, needs a specific one), and for sentences lines 154-6 and 198-201, and protection in upper respiratory airways line 371-onward. The topic is very timely and important, the review is too low quality as it is and needs major improvements to be of any significance. The recent UK biobank report adds to our knowledge in real time, these reports should be added. Rapid publication of a revised version should be warranted.

Author Response

The subject is very timely for a review, however, the authors fail to address some key issues. To name a few:

Epitope mapping and HLA restriction of antibody responses to SARS-CoV2; the presence of natural antibodies in non-infected subjects; local and systemic mucosal responses and their memory (oral boosters); abnormal antibodies (see a-fucosylated IgG, their binding to activating monocyte Fc receptors); and finally, poor type I IFN responses and/or presence of antibodies to IFN type I. The coverage of vaccine trials and their published correlates is adequate. The use of possible standardized tests is not convincingly advanced, no examples are offered. There are problems with the English language (e.g. lines 190. 305 (an), 343 and following 420). Some abbreviations are not explained (nAB line 110, then newly in line 416, are they the same?), NHP line 338, 

Missing references needed for HepB (line 101), measles (ref 17 is generic, needs a specific one), and for sentences lines 154-6 and 198-201, and protection in upper respiratory airways line 371-onward.

The topic is very timely and important, the review is too low quality as it is and needs major improvements to be of any significance.

The recent UK biobank report adds to our knowledge in real time, these reports should be added. Rapid publication of a revised version should be warranted.

Reply: We would like to thank the reviewer for the overall positive feedback and the constructive and critical reading of the review. We addressed the concerns raised during the review process. We believe that these modifications have further strengthened our manuscript.

We included the aspects of local and systemic mucosal responses and abnormal antibodies as well as type I IFN responses. Additionally, we mentioned specific assays to evaluate humoral (ELISA against S and RBD, surrogate assays and VNT assays) and cell-mediated immune responses (ELISpot). Furthermore, we included information on different study designs to identify CoP and provided specific examples. We also added the reference of the UK Biobank report in our manuscript.

Regarding the aspects of natural antibodies in non-infected subjects, cross-reactive antibodies have been reported in patients who survived SARS-CoV-1, although the detectable antibodies were largely non-neutralizing[1]. Cross-reactivity with other seasonal human CoVs is an active area of research, but few peer-reviewed data are available to date. If cross-reactive antibodies would have significant cross-neutralizing activities, this could explain phenomena like the neutralizing capacity of regular intravenous immunoglobulin[2]. However, while there are recent studies showing T-cell reactivity against a number of SARS-CoV-2 epitopes in people with no known exposure to the virus[3-6], validated data on the cross-reactive antibodies is still scarce. We have therefore decided not to address this aspect in our review.

Further, we thank the reviewer for the aspect of epitope mapping and HLA restriction and agree that this is a valuable information. Nevertheless, we focused our review on study designs and CoP and more detailed information on specific immune responses and specific assays would be out of the scope of our manuscript. While epitope mapping and HLA typing can provide critical impact on disease severity, we would like to concentrate on vaccine-induced CoP and the standardized assays like ELISA and VNTs.

We modified the manuscript and corrected the English language, included explanations of abbreviations and added the missing references.

In summary, we thank the reviewer for thoroughly reviewing our manuscript and believe that the comments we received helped us to substantially improve the Review.

  1. Lv H, Wu NC, Tsang OT-Y, Yuan M, Perera RA, Leung WS, et al. Cross-reactive antibody response between SARS-CoV-2 and SARS-CoV infections. Cell reports. 2020;31(9):107725.
  2. Xie Y, Cao S, Li Q, Chen E, Dong H, Zhang W, et al. Effect of regular intravenous immunoglobulin therapy on prognosis of severe pneumonia in patients with COVID-19. The Journal of infection. 2020.
  3. Grifoni A, Weiskopf D, Ramirez SI, Mateus J, Dan JM, Moderbacher CR, et al. Targets of T cell responses to SARS-CoV-2 coronavirus in humans with COVID-19 disease and unexposed individuals. Cell. 2020;181(7):1489-501. e15.
  4. Weiskopf D, Schmitz KS, Raadsen MP, Grifoni A, Okba NM, Endeman H, et al. Phenotype and kinetics of SARS-CoV-2-specific T cells in COVID-19 patients with acute respiratory distress syndrome. Sci Immunol. 2020;5(48).
  5. Le Bert N, Tan AT, Kunasegaran K, Tham CY, Hafezi M, Chia A, et al. SARS-CoV-2-specific T cell immunity in cases of COVID-19 and SARS, and uninfected controls. Nature. 2020;584(7821):457-62.
  6. Meckiff BJ, Ramírez-Suástegui C, Fajardo V, Chee SJ, Kusnadi A, Simon H, et al. Single-cell transcriptomic analysis of SARS-CoV-2 reactive CD4+ T cells. Available at SSRN 3641939. 2020.

Reviewer 2 Report

In this review, the authors discuss correlates of vaccine-induced protection (CoP) against SARS-CoV-2. Authors compare immune markers/responses against natural SARS-CoV2 infection versus vaccine-induced immune responses associated with protection . The review points out that CoP could be an important measure for evaluating novel vaccine candidates and their efficacy.  

The topic is very interesting, and authors cover a unique aspect for measuring the protection against SARS-CoV-2. Overall, the manuscript is well written and clearly presented, unless otherwise indicated below.

Authors can be invited to revise the following points before the publication of this work. 

(1). Since authors emphasize that randomized placebo-controlled trials might not be appropriate anymore. Therefore, correlates of protection (CoP) could be an important measure for evaluating novel vaccine candidates. It is imperative to discuss more randomized placebo-controlled trials, with both aspects in which scenario they are appropriate and in which case they are not appropriate. i.e. provide a balanced view.

Then discuss correlates of protection (CoP) compared to randomized placebo-controlled trials.

Authors may also wish to discuss Correlates of Protection Induced by Vaccination (PMID: 20463105). 

(2). While it has been stated that ‘’An immune response that is responsible for and statistically interrelated with protection is named correlate of protection (CoP)’’. Author may also mention these immune responses measured by signs?, immune markers/or antibodies. Perhaps more specifically signs and immune markers against covid19. 

(3). Page 2, line # 85, 86 (of pdf file): Authors have indicated that CoP might be useful to measure vaccine efficacy ‘’without the need to measure clinical outcome’’. Since the CoP is taken into consideration for measuring immune response/immune markers in the sense of being protected against becoming infected (i.e associated with protection against infection or disease), as well as evaluating population immunity, how clinical outcomes can be ignored?

Of course, vaccine efficacy can be measured without the need to measure clinical outcome, however by definition markers in the sense of being protected against becoming infected and/or developing disease, the clinical outcome is important. 

(4). Figure 1: please expand/define (write at full) the abbreviations in figure legends. Also please label each cell type. In addition to their names in legends, please also label the cells themselves antigen presenting cells/dendritic cells, B cells, T cells, and antibodies. 5) Uptake of APC by? 5) uptake of x by y.

(5). Page 4 line 106: while authors mention ‘’Good examples to delineate the complexity of identifying CoP are measles and poxviruses vaccines’’, the following article can also be discussed.

PMID: 31674648 and PMID: 19268607

Then also correlate it with SARS-CoV2 (PMID: 32875286 and PMID: 33276369). 

(6). Finally, central to current review which is CoP of vaccine-induced protection against SARS-CoV-2. How CoP for natural infection can guide vaccine development and efficacy.

For example, when immune responses (antibodies) against covid19, responsible for protection (CoP) are considered, it is necessary to discuss the ‘’time window’’ for detection of natural or vaccine induced immune responses related to SARS-CoV2 protection. What ‘’time window’’ is related with natural CoP?, and at what time window the CoP is not applicable?. If we consider IgG, can CoP be applied only from 20 to 180 days or so? (In the light of figure 2). How this CoP can guide vaccine development or protection and vaccine efficacy. Discuss it. 

Other typos: 

  • Abstract: In this Review → In this review
  • we will report → we report
  • immunogenicity to natural SARS-CoV-2 infection → immunogenicity against natural SARS-CoV-2 infection.
  • Page 4, line 116: In this review, we will delineate → In this review, we delineate

Author Response

Comments and Suggestions for Authors

In this review, the authors discuss correlates of vaccine-induced protection (CoP) against SARS-CoV-2. Authors compare immune markers/responses against natural SARS-CoV2 infection versus vaccine-induced immune responses associated with protection. The review points out that CoP could be an important measure for evaluating novel vaccine candidates and their efficacy.  

The topic is very interesting, and authors cover a unique aspect for measuring the protection against SARS-CoV-2. Overall, the manuscript is well written and clearly presented, unless otherwise indicated below.

Reply: We would like to thank the reviewer for the positive feedback and the constructive and critical reading of the review. We replied to each comment and modified the review to reflect the suggestions.

Authors can be invited to revise the following points before the publication of this work. 

(1). Since authors emphasize that randomized placebo-controlled trials might not be appropriate anymore. Therefore, correlates of protection (CoP) could be an important measure for evaluating novel vaccine candidates. It is imperative to discuss more randomized placebo-controlled trials, with both aspects in which scenario they are appropriate and in which case they are not appropriate. i.e. provide a balanced view.

Then discuss correlates of protection (CoP) compared to randomized placebo-controlled trials.

Authors may also wish to discuss Correlates of Protection Induced by Vaccination (PMID: 20463105). 

Reply: We thank the reviewer for their comment and agree that the aspect of vaccine-induced CoP needed to be more highlighted and better cited. We have added the suggested literature, re-structured our manuscript and added information on how to identify CoP using specific study designs. We believe that these modifications have further strengthened the key aspect of the review.   

(2). While it has been stated that ‘’An immune response that is responsible for and statistically interrelated with protection is named correlate of protection (CoP)’’. Author may also mention these immune responses measured by signs? immune markers/or antibodies. Perhaps more specifically signs and immune markers against covid19. 

Reply: We appreciate the reviewers input and agree that immune markers or other measurements that predict course of disease of COVID-19 are an interesting topic. The measurement of the neutrophil to lymphocyte ration (NLR) might be an example of a readily available measurement to predict outcome in COVID-19 (https://www.springermedizin.de/covid-19/predictive-values-of-neutrophil-to-lymphocyte-ratio-on-disease-s/18590612). Other immunological markers associated with severe disease which might not be as easily measurable in clinical routine include NLRP3, IL-6, IL-12 and IL-1β. Although the topic of immune markers predicting disease severity is very important , valuable and interesting, in our opinion this aspect is beyond the scope of this review; and therefore we have decided not to include it in the manuscript.

(3). Page 2, line # 85, 86 (of pdf file): Authors have indicated that CoP might be useful to measure vaccine efficacy ‘’without the need to measure clinical outcome’’. Since the CoP is taken into consideration for measuring immune response/immune markers in the sense of being protected against becoming infected (i.e associated with protection against infection or disease), as well as evaluating population immunity, how clinical outcomes can be ignored? Of course, vaccine efficacy can be measured without the need to measure clinical outcome, however by definition markers in the sense of being protected against becoming infected and/or developing disease, the clinical outcome is important. 

Reply: We thank the reviewer for this valuable comment and agree that the wording needed to be revised as it is not clearly understandable. We have now rephrased the sentence as follows: “An immune response that is responsible for and statistically interrelated with protection is named correlate of protection (CoP). A CoP would allow to predict clinical outcomes (protection against disease or infection) after vaccination or following natural infection.” In addition, we re-structured the manuscript and believe that we now transfer a better understanding of evaluating CoP by describing potential study designs that enable the identification of CoP. We believe that the manuscript now better describes the benefits and challenges and limitations of using CoP.

(4). Figure 1: please expand/define (write at full) the abbreviations in figure legends. Also please label each cell type. In addition to their names in legends, please also label the cells themselves antigen presenting cells/dendritic cells, B cells, T cells, and antibodies. 5) Uptake of APC by? 5) uptake of x by y.

Reply: We thank the reviewer for the comment and have modified the figure accordingly.

(5). Page 4 line 106: while authors mention ‘’Good examples to delineate the complexity of identifying CoP are measles and poxviruses vaccines’’, the following article can also be discussed.

PMID: 31674648 and PMID: 19268607

Then also correlate it with SARS-CoV2 (PMID: 32875286 and PMID: 33276369).  

For example, when immune responses (antibodies) against covid19, responsible for protection (CoP) are considered, it is necessary to discuss the ‘’time window’’ for detection of natural or vaccine induced immune responses related to SARS-CoV2 protection. What ‘’time window’’ is related with natural CoP?, and at what time window the CoP is not applicable?. If we consider IgG, can CoP be applied only from 20 to 180 days or so? (In the light of figure 2). How this CoP can guide vaccine development or protection and vaccine efficacy. Discuss it.  

Reply: We agree that measurement of CoP needs to be timed critically. However, we find that evidence for reasonable recommendations is lacking. E.g. kinetics of cellular immune response as well as antibody kinetics in patients that receive a delayed second dose remain to be investigated. We have discussed these aspects in the manuscript.

Other typos: 

  • Abstract: In this Review → In this review
  • we will report → we report
  • immunogenicity to natural SARS-CoV-2 infection → immunogenicity against natural SARS-CoV-2 infection.
  • Page 4, line 116: In this review, we will delineate → In this review, we delineate

Reply: We thank the reviewer for pointing out the typos and have corrected the mistakes.

Round 2

Reviewer 1 Report

Despite strenuous efforts and improvements, parts of this article are still unsatisfactory. Lines 210-13 have unclear statements or wrong ones; line 219 puts mRNA; including the AstraZeneca vaccine which is an adeno-based one; line 293 assumes identity of sterilizing and mucosal immunity, which is NOT; line 178 includen IFN gamma among type I IFN, a gross mistake even for a student; ref 66 is still inadequate for measles (there are tons of papers on CoP in measles infection and vaccines); the same ref 66 is used for poxviruses (authors cite a book because they are not familiar with these vaccinations?); line 373 contains 2 errors: GrzB is not expklained, and Tfh cells for protection needs a reference; line 452 BBIPP is not explained; 469 and 497 contain misspellings or mistakes, and so on and on. The kanguage still needs improvements and clarification.

Author Response

Despite strenuous efforts and improvements, parts of this article are still unsatisfactory.  

Reply: We would like to thank the reviewer for the critical reading of the revised manuscript. We addressed the concerns raised during the second review process. We believe that these modifications have further strengthened our manuscript.

Lines 210-13 have unclear statements or wrong ones;

Reply: We thank the reviewer for this valuable comment and agree that the wording needed to be revised as it is not clearly understandable. We have modified the paragraph as follows:

In conclusion, several factors such as age, obesity, or sex can influence the risk to develop severe COVID-19[58]. Coordinated T-cell and antibody responses appear to be protective, while uncoordinated responses fail to control or even promote disease. 

line 219 puts mRNA; including the AstraZeneca vaccine which is an adeno-based one;

Reply: We thank the reviewer for pointing out this mistake. We have corrected the manuscript and deleted the word mRNA.

line 293 assumes identity of sterilizing and mucosal immunity, which is NOT;

Reply: We appreciate the input and agree that our wording was not clear and definitely needs to be revised. We modified the wording of the paragraph and stated the difference between sterilizing and mucosal immunity more clearly as follows:

Further vaccine candidates are expected to be approved soon, most of them will be administered intramuscularly. While those generally induce systemic immune responses with dominant IgG responses, natural infection induces both systemic and mucosal immune responses[66, 67]. The induction of mucosal immune response in the upper respiratory tract generally leads to secretion of secretory IgA, which can be an important factor to induce sterilizing immunity preventing infection and virus transmission[67]. A vaccine candidate that induces mucosal immune response in the upper respiratory tract and thereby potentially sterilizing immunity would be preferable. It has been shown that e.g. application of viral vectors intranasally can lead to strong mucosal immune responses as well as an IgG response[67]. To date, six intranasal and three oral vaccine candidates are in clinical Phase 1 or 2 trials [2]. While data from clinical trials have not yet been published, preclinical studies suggest the induction of mucosal immunity [68]. First results from clinical trials of an oral vaccine candidate by Vaxart Inc. have recently been announced[69].

line 178 includen IFN gamma among type I IFN, a gross mistake even for a student;

Reply: We thank the reviewer for this valuable comment and agree that we made a mistake in our writing process. We corrected our revised version as follows:

“T-cellular immune response to viral infection is mediated by interferon (IFN) with a major role of type I IFN[54]. IFN-stimulated genes (ISGs) have been shown to be significantly reduced in critical COVID-19 patients compared with patients that experienced mild to moderate infection[55].”

ref 66 is still inadequate for measles (there are tons of papers on CoP in measles infection and vaccines); the same ref 66 is used for poxviruses (authors cite a book because they are not familiar with these vaccinations?);

Reply: We have added additional references.

line 373 contains 2 errors: GrzB is not expklained, and Tfh cells for protection needs a reference;

Reply: We thank the reviewer for the comment. We have added granzyme B (GrzB) and included a reference for Tfh (Holmgren et al. Correlates of protection for enteric vaccines. Vaccine. 2017;35(26):3355-63.). In the cited study, Holmgren and colleagues describe how Tfh modulate the immune response to vaccines, e.g. by inducing vaccine-specific IgA production.

line 452 BBIPP is not explained;

Reply: We thank the reviewer for the comment. We have modified the sentence to now consistently state the company and not the name of the vaccine. We have deleted BBIP and replaced it with Sinopharm.

 469 and 497 contain misspellings or mistakes, and so on and on.

 Reply: We thank the reviewer for pointing out the misspellings and have corrected the mistakes. We have modified several sentences in the manuscript.

 The kanguage still needs improvements and clarification.

 Reply: We have revised the manuscript regarding the language.

Reviewer 2 Report

Authors have made significant improvements in their manuscript after addressing the reviewer comments, and adding new parts in the text. 

The message is conveyed.  

I endorse the publication of this work. 

Author Response

Authors have made significant improvements in their manuscript after addressing the reviewer comments, and adding new parts in the text. 

The message is conveyed.  

I endorse the publication of this work. 

Reply: We would like to thank the reviewer for the critical reading of the revised manuscript and the overall positive feedback.

Round 3

Reviewer 1 Report

great improvements